# Evaluation of Serum and Salivary Iron and Ferritin Levels in Children with Dental Caries: A Meta-Analysis and Trial Sequential Analysis

**DOI:** 10.3390/children8111034

**Published:** 2021-11-11

**Authors:** Roohollah Sharifi, Mohammad Farid Tabarzadi, Parsia Choubsaz, Masoud Sadeghi, Jyothi Tadakamadla, Serge Brand, Dena Sadeghi-Bahmani

**Affiliations:** 1Department of Endodontics, School of Dentistry, Kermanshah University of Medical Sciences, Kermanshah 6713954658, Iran; roholahsharifi@gmail.com; 2Students Research Committee, Kermanshah University of Medical Sciences, Kermanshah 6715847141, Iran; carlinacross@gmail.com; 3Department of Orthodontics, School of Dentistry, Shahid Beheshti University of Medical Sciences, Tehran 1983963113, Iran; parsia.choubsaz@sbmu.ac.ir; 4Department of Biology, Science and Research Branch, Islamic Azad University, Tehran 1477893855, Iran; sadeghi_mbrc@yahoo.com; 5School of Medicine and Dentistry, Griffith University, Brisbane, QLD 4222, Australia; j.tadakamadla@griffith.edu.au; 6Sleep Disorders Research Center, Kermanshah University of Medical Sciences, Kermanshah 6719851115, Iran; bahmanid@stanford.edu; 7Center for Affective, Stress and Sleep Disorders, University of Basel, Psychiatric Clinics, 4001 Basel, Switzerland; 8Substance Abuse Prevention Research Center, Kermanshah University of Medical Sciences, Kermanshah 6715847141, Iran; 9Department of Sport, Exercise and Health, Division of Sport Science and Psychosocial Health, University of Basel, 4052 Basel, Switzerland; 10School of Medicine, Tehran University of Medical Sciences, Tehran 1416753955, Iran; 11Department of Psychology, Stanford University, Stanford, CA 94305, USA

**Keywords:** dental caries, serum, saliva, iron, ferritin, meta-analysis

## Abstract

Background and objective: Dental caries appears to be related to iron deficiency anemia and to low ferritin levels. In the present meta-analysis, we report salivary and serum iron and ferritin levels in children with dental caries, compared to healthy controls. Materials and methods: We searched in Web of Science, Cochrane Library, Scopus, and PubMed/Medline databases to extract studies published until 25 July 2021. We calculated mean differences (MD) and 95% confidence intervals (CI) of salivary and serum iron and ferritin levels in children with dental caries, always compared to healthy controls. In addition, we applied a trial sequential analysis (TSA). Results: A total of twelve articles covering thirteen studies were included in the meta-analysis. The pooled MD for salivary iron level was −5.76 µg/dL (*p* = 0.57), and −27.70 µg/dL (*p* < 0.00001) for serum iron level: compared to healthy controls, children with dental caries did not show different salivary iron levels, while children with caries had significantly lower serum iron levels. The pooled MD of salivary ferritin level was 34.84 µg/dL (*p* = 0.28), and the pooled MD of serum ferritin level was −8.95 µg/L (*p* = 0.04): compared to healthy controls, children with dental caries did not have different salivary iron levels, but significantly lower serum ferritin levels. Conclusions: The findings of the present meta-analysis showed that salivary levels of iron and ferritin did not differ between children with and without caries, though compared to healthy controls, children with caries had significantly lower salivary and serum iron and ferritin levels. The results are of practical and clinical importance: Possibly, iron and ferritin supplementation might prevent or attenuate dental caries in children at risk. Further, children with caries might suffer from further iron- and ferritin-related health issues. Lastly, serum blood samples, but not saliva samples inform accurately about the current iron and ferritin concentrations in children with or without caries.

## 1. Introduction

Dental caries is caused by the chemical dissolution of tooth tissues as a result of acids produced by bacteria that metabolize carbohydrates from the diet, especially sucrose [1]. The World Health Organization (WHO) reported dental caries as a pandemic disease impacting 60–90% of school-aged children [2]. Thus, dental caries is a major problem that could have a debilitating impact on children, their families and also health system [3]. Causes of caries in children include poor nutrition, constant exposure to sugar-rich foods and the presence of cariogenic microorganisms in the oral cavity [4]. As regards adolescents, biological, social, behavioral, psychological and economic conditions appeared to strongly impact on the emergence and maintenance of dental caries [5]. Saliva plays a protective role in dental caries because it lubricates the oral cavity, has antibacterial properties, rinses the oral cavity and removes food debris, and also has a buffering action that involves the activity of bicarbonate ions [6]. Therefore, dental caries has a multifactorial etiology and its further development depends on the interaction of three major factors: microbial organisms, substrate, and the host (teeth and saliva) [7].

Dental caries is related to iron deficiency anemia and to lowered serum ferritin levels [8]. Importantly, a bidirectional process can be observed: dental caries, its discomfort and pain can interfere with proper nutrition, including iron intake. A low iron intake can lead to iron deficiency anemia [9,10]; as a result, such an iron and ferritin deficiency appears to further unfavorably impact on the development of caries. Further, a potential inhibitory mechanism of iron on dental caries due to the inhibition of cariogenic bacteria has also been proposed [11,12]. Iron deficiency is the most common nutritional deficiency in childhood, often accompanied by severe caries damage [13]. Furthermore, iron deficiency can affect a child’s physical and mental development and is generally characterized by low levels of hemoglobin and/or ferritin [14]. Despite the high prevalence of dental caries and iron deficiency, limited studies are available on the relationship between the two factors [3]. Ferritin is an iron-storage protein that releases iron for essential cellular processes [15]. Low levels of ferritin indicate that the body has reduced iron stores to keep hemoglobin at a healthy level [16]. As mentioned above, iron and ferritin deficits reflect a major health issue in general and a risk of developing caries, in specific, and this holds particularly true for children. With this background in mind, and given that to our knowledge, there is no meta-analysis evaluating the levels of iron and ferritin among children with and without dental caries the aim of the present meta-analysis was to calculate, if salivary and serum iron and ferritin levels did systematically differ between children with and without caries. If true, then an appropriate treatment might attenuate the risk of both caries and iron- and ferritin-related health issues.

## 2. Materials and Methods

### 2.1. Design Protocol

The PRISMA guidelines were applied to report this meta-analysis [17]. The PICO (patient/population, intervention, comparison, and outcomes) question was: do salivary and serum iron and ferritin levels differ between children with and without dental caries?

### 2.2. Search Strategy

We selected four databases; the Web of Science, Cochrane Library, Scopus, and PubMed/Medline, to search for studies published until 25 July 2021, without any restrictions (language, publication year, age, sex, and etc.). The search was adapted to the databases, search used in PubMed was: (“caries” or “decay*”) and (“iron” or “Fe” or “Fe^2+^” or “ferritin” or “metal*” or “element*”) and (“serum” or “blood” OR “saliva*”) and (“control*” or “caries-free” or “caries free” or “non-carious”). In addition, we checked the citations of articles related to our topic to extract potential articles.

### 2.3. Criteria

M.S. read the titles and abstracts of all studies retrieved from the databases and checked them for duplicates and irrelevant studies. A study was relevant if the study met the following inclusion criteria: (1) case-control, cohort, or cross-sectional designs; (2) dental caries was the outcome of interest; (3) reporting salivary and/or serum levels of iron or ferritin; (4) reporting the required data to calculate the mean differences (MDs) with 95% confidence intervals (CIs); (5) including age group of less than 18 years, and (6) including participants with no caries as control group. Exclusion criteria were: (1) animal studies; (2) meta-analyses; (3) reviews; (4) letter to the editors; (5) reports of previous studies; (6) commentaries to and corrections of the previous studies. The second author (R.S) re-checked the relevant articles based on the inclusion and exclusion criteria. In the event of disagreement between, a third author was involved (S.B).

### 2.4. Data Extraction

Two authors (M.S; M.F.T) independently extracted the data from each study and compared the results. The following information were gathered: first author’s name; publication year; sample size, range and mean age, and male-female-ratio among children with and without dental caries, and the quality score (see below). In the case of low consensus, a third author (J.T) helped to solve the issue.

### 2.5. Quality of Assessment

One author (M.S) identified the quality of each included study using the Newcastle–Ottawa scale (NOS) questionnaire with three domains (selection (0–4 points), comparability (0–2 points), and exposure (0–3 points)); the maximum possible score for each study was nine points [18]. This quality assessment scale involves three domains (maximum stars): selection (four stars), comparability (two stars), and exposure (three stars). Each star reflects one point.

### 2.6. Statistical Analysis

We used Review Manager 5.3 (RevMan 5.3) to perform the meta-analysis. The effect estimate, mean difference (MD) and 95% confidence interval (CI) were calculated for each study to demonstrate the difference in salivary and serum iron and ferritin levels between children with and without dental caries. To evaluate the pooled MD significance, the Z-test was applied; a *p*-value less than 0.05 was considered as significant. The Cochrane Q-test and I^2^ statistic were utilized to assess the heterogeneity. If there was a statistically significant heterogeneity (*p* < 0.1 or I^2^ > 50%), we used a random-effect model [19]; otherwise, the fixed-effect model [20] was used.

The funnel plots were analyzed by the Egger’s and Begg’s tests, and a *p* < 0.05 demonstrates statistically significant existence of the publication bias. To evaluate the stability of the pooled results, sensitivity analyses (“one study removed” and “cumulative analysis”) were conducted. These analyses were performed by the Comprehensive Meta-Analysis version 2.0 (CMA 2.0).

Each meta-analysis may create a false-positive or negative conclusion [21]. Given this, trial sequential analysis (TSA) was conducted using TSA software (version 0.9.5.10 beta) (Copenhagen Trial Unit, Centre for Clinical Intervention Research, Rigs Hospitalet, Copenhagen, Denmark) to reduce these statistical errors [22]. Additionally, a threshold of futility could be tested by TSA to reach a conclusion of no effect before reaching the information size. We estimated D^2^ as 100% for the salivary iron and 89% for the serum ferritin levels, and the mean differences and variance based on empirical assumptions that were autogenerated by the software. The sample size is considered sufficient when the Z-curve reaches the required information size (RIS) line, the boundary line or futility area, and the results were considered conclusive. Otherwise, the amount of information available was considered to be inadequate to make valid conclusions, and more evidence is needed. The unit of measurement for the salivary iron and ferritin levels was µg/dL; the unit of measurement for serum iron and ferritin was µg/L.

## 3. Results

### 3.1. Study Selection

A total of 485 records were identified (Figure 1). After removing duplicates and irrelevant records, 24 full-text articles were considered for inclusion. After assessing the articles in full, twelve were excluded for the following reasons: one article was a review; five articles had no control groups; one article had no sufficient data; one article did not report the number of cases and controls; three articles included adults, and one article included a control group with low caries. A total of 12 articles reporting thirteen studies were entered into the meta-analysis.

### 3.2. Characteristics of the Studies

Table 1 shows the characteristics of 12 studies [3,16,23,24,25,26,27,28,29,30,31,32] included in the meta-analysis. The studies were published between 2011 and 2021. Five studies reported salivary iron levels [23,24,25,28,32], one [31] reported both serum iron and ferritin levels, two [27,29] reported salivary ferritin levels, and four [3,16,26,30] reported serum ferritin levels.

### 3.3. Quality Assessment

Table 2 is shown the quality score of each study based on the NOS. All studies had NOS quality scores of ≥7.

### 3.4. Pooled Analysis

Figure 2 shows salivary (five studies) and serum iron levels (one study) in children with dental caries, compared to children without dental caries. For salivary iron levels the pooled MD was −5.76 µg/dL (95%CI: −25.69, 14.18; *p* = 0.57; I^2^ = 99%), that is to say: salivary iron levels did not statistically differ between children with and without dental caries. For serum iron levels, the MD was −27.70 µg/dL (95%CI: −35.63, −19.77; *p* < 0.00001), that is to say: compared to children without dental caries, children with dental caries had statistically significantly lower serum iron levels.

Figure 3 shows the pooled analysis of salivary (two studies) and serum ferritin levels (five studies) in children with dental caries, compared to children without dental caries. For salivary ferritin levels, the pooled MD was 34.84 µg/dL (95%CI: −28.78, 98.46; *p* = 0.28; I^2^ = 100%); that is to say: salivary ferritin levels did not statistically differ between children with and without dental caries. For salivary ferritin levels the pooled MD was −8.95 µg/L (95%CI: −17.61, −0.29; *p* = 0.04; I^2^ = 87%), that is to say: serum ferritin levels were statistically significantly lowered among children with dental caries, compared to children without dental caries.

### 3.5. Sensitivity Analysis

Both sensitivity analyses (“one-study-removed” and “cumulative analyses”) were performed on the salivary iron and serum ferritin levels; results indicated stability of the initial results.

### 3.6. Publication Bias

The funnel plots for publication bias among studies reporting salivary iron and serum ferritin levels are presented in Figure 4. Both funnel plots and the statistical tests did not reveal any publication bias for salivary iron and serum ferritin levels (Egger’s: 0.204 and 0.356 and Begg’s: 1.000 and 0.327, respectively)

### 3.7. Trial Sequential Analysis (TSA)

Figure 5 shows TSA of studies involving salivary iron and serum ferritin levels. The Z-curve did not reach the RIS line or cross the boundary line or enter futility area, establishing that the studies were not enough, and more studies was needed in the future to confirm or reject the results.

## 4. Discussion

The findings of the present meta-analysis showed that salivary iron and ferritin levels did not statistically significantly differ between children with and without dental caries. In contrast, compared to children with no dental caries, children with dental caries had statistically significantly lower serum iron and ferritin levels. While the pattern of results was satisfactorily robust, we also note that the number of studies was modest.

More specifically, as regards salivary iron levels, three [23,25,32] out of five studies showed that salivary iron levels and dental caries were unrelated (one of the studies [23] was a pilot study); by contrast, one study [28] found higher salivary iron levels among children with caries, compared to children without caries; however, another study found significantly higher salivary iron levels among children with no caries, compared to those with caries [24].

As regards serum iron levels, one study [31] reported lower serum iron levels in children with caries, compared to children without caries.

As regards serum ferritin levels, three out of five studies [16,26,31] reported statistically significantly lower serum ferritin levels in children with caries, compared to children without caries. By contrast, two out of five studies [3,30] could not identify statistically significantly serum ferritin levels differences between children with or without dental caries. Last, two out of five studies [27,29] reported statistically significantly higher salivary ferritin levels in children with caries, compared to children without caries.

One study claimed that severe early childhood caries may be a risk marker for iron deficiency anemia; additionally, iron deficiency may permanently negatively impact on a child’s growth and development [33]. To prevent and control the development of caries in children, and thus to enable a healthy growth and development, it appears mandatory to thoroughly supervise children’s dietary intake in general, and glucose intake (“sugar”) in specific [34,35]. Dental caries is associated with an increased inflammatory response and a subsequent decrease of hemoglobin levels. As such, the production of cytokines is decreased, which in turn further blocks the erythropoiesis [36].

The present meta-analysis revealed a broad range and heterogeneity of iron and ferritin levels among children with and without dental caries. The quality of the single studies did not always allow a deepened understanding of the underlying mechanisms and of possible further, but unassessed factors.

Here, we mention the following possible confounders: age, sex, assessment of iron and ferritin levels, and BMI. In this regard, sex and age may also influence ferritin and iron levels [37].

In regard to the present meta-analysis, participants’ age varied across the single studies; further, not all studies reported thoroughly participants’ age range and gender-ratio. Similarly, assessments of iron and ferritin levels varied across the studies, which might have had blurred the whole pattern of results. Next, higher BMI indices were associated with lower serum ferritin levels [38]. To understand this association, it appears conceivable that lower BMI indices might be associated with unbalanced eating habits with too high carbohydrate/sugar intake; as a result, children with lower BMI might be at higher risk of developing dental caries [39]. We note that the studies included in the present meta-analysis did not report participants’ BMI indices; as such, BMI indices could not be entered in the calculations as a possible confounder. Overall, future studies on the associations between iron and ferritin levels and dental caries should consider reporting age, sex, and BMI both in children with and without dental caries as possible confounders.

To quality of the studies does not allow to ultimately explain the different correlation coefficients between the occurrence of caries and iron and ferritin in saliva and serum. We speculate that the low number of published studies, different lab kits to measure iron and ferritin, and the large range of sample sizes might have conferred to this blurred pattern of results. In this view, and as the TSA showed, more studies with higher numbers of cases are needed to observe more robust results. However, at this stage and with this low number of studies, to diagnose the prevalence and progression of dental caries, it appears more reliable to assess serum iron and ferritin levels in serum, but not in saliva.

Limitations of the present meta-analysis were: (1) the modest number of studies precluded both sub-group and meta-regression analyses; (2) exclusively published studies were taken into consideration; (3) there was high heterogeneity between the studies as regards the number of participants, and the quality of assessment to measuring both iron and ferritin levels in saliva and in blood serum. (4) A partial or complete lack of participants’ age, sex, and BMI. (5) The present study protocol was not registered in databases such as *PROSPERO.* In contrast, strengths were the lack of publication bias and the overall high quality of the studies.

## 5. Conclusions

The findings of the present meta-analysis showed that salivary iron and ferritin levels did not differ between children with and without dental caries. By contrast, and compared to healthy controls, children with dental caries had significantly lower serum iron and ferritin levels. Given this, children with dental caries might bear the risk of further iron- and ferritin-related health issues. However, to ascertain possible iron and ferritin deficits, blood serum samples, but not saliva samples, are mandatory. Keeping in mind the modest number of studies, we are cautious to generalize the present results. However, we suggest that compared to saliva, serum iron and ferritin levels appear more reliable biomarkers to diagnose the prevalence and disease progression of caries in children.

## Figures and Tables

**Figure 1 children-08-01034-f001:**
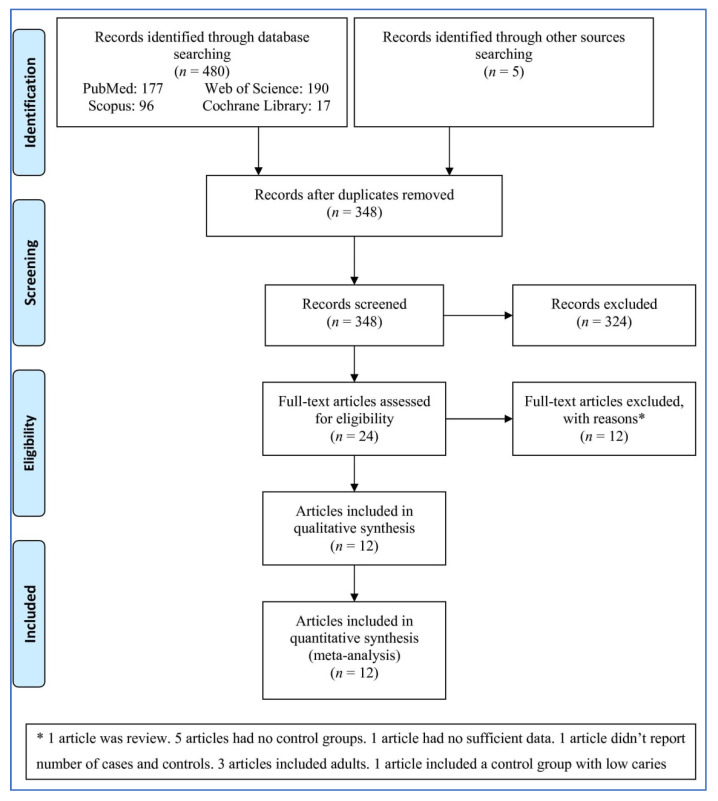
Flowchart demonstrating the study selection procedure. “*” means the reasons of excluding of the full-text articles.

**Figure 2 children-08-01034-f002:**
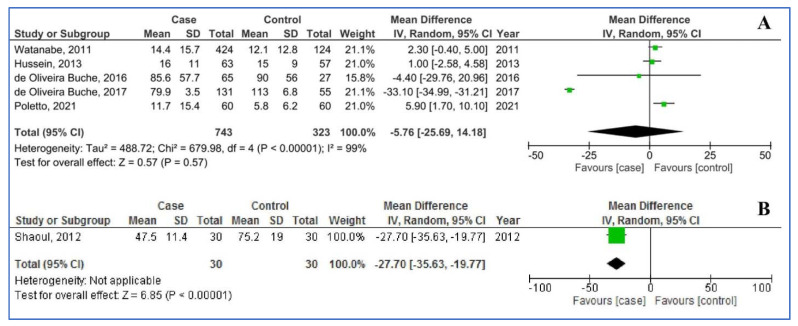
Pooled analysis of salivary iron (**A**) and serum iron (**B**) levels in the participants with dental caries (cases) compared to those without caries (controls). Abbreviations: SD, Standard Deviation; IV, Inverse Variance; CI, Confidence Interval.

**Figure 3 children-08-01034-f003:**
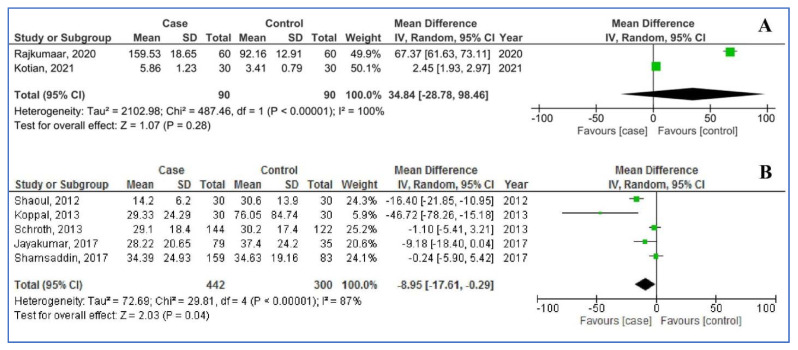
Pooled analysis of salivary ferritin (**A**) and serum ferritin (**B**) levels in the participants with dental caries (cases) compared to free caries (controls).

**Figure 4 children-08-01034-f004:**
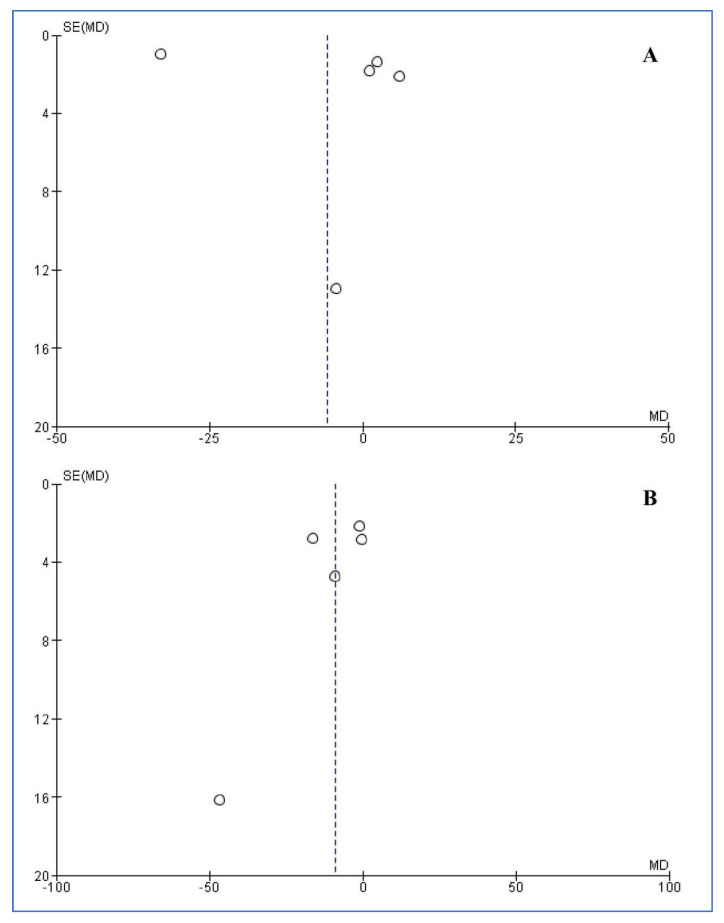
Funnel plots of included studies reporting salivary iron (**A**) and serum ferritin (**B**) levels.

**Figure 5 children-08-01034-f005:**
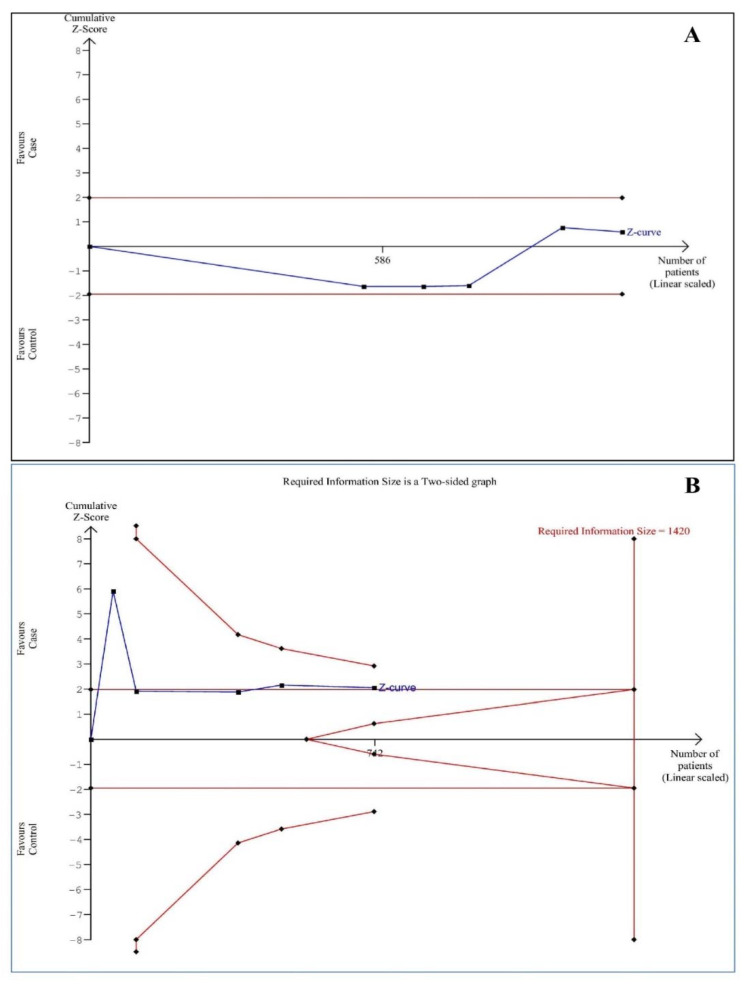
Trial sequential analysis of studies assessing salivary iron (**A**) and serum ferritin (**B**) levels.

**Table 1 children-08-01034-t001:** Characteristics of the studies included in the meta-analysis.

First Author, Publication Year	Country	Number of Cases to Controls	Age Range of All Participants (Mean Age of Cases to Controls)	Male/Female of Cases to Controls	Outcome Evaluated	Measurement Method
Watanabe, 2011 [32]	Japan	424 to 124	6–12	NR	Salivary iron	AAS
Shaoul, 2012 [31]	Israel	30 to 30	3–18 (5.7 to 5.8)	17/13 to 16/14	Serum iron and ferritin	NR
Hussein, 2013 [25]	Malaysia	63 to 57	8–12	NR	Salivary iron	AAS
Koppal, 2013 [16]	India	30 to 30	2–6	NR	Serum ferritin	NR
Schroth, 2013 [30]	Canada	144 to 122	<6	NR	Serum ferritin	NR
de Oliveira Buche, 2016 [23]	Brazil	65 to 27	11–14 (12 to 12)	NR	Salivary iron	Colorimetric test
de Oliveira Buche, 2017 [24]	Brazil	131 to 55	11–14	NR	Salivary iron	Colorimetric test
Jayakumar, 2017 [26]	India	79 to 35	<6	NR	Serum ferritin	ECLIA
Shamsaddin, 2017 [3]	Iran	159 to 83	2–6	NR	Serum ferritin	NR
Rajkumaar, 2020 [29]	India	60 to 60	4–4 (4.2–4.1)	34/26 to 32/28	Salivary ferritin	CMIA
Kotian, 2021 [27]	India	30 to 30	3–6	NR	Salivary ferritin	CMIA
Poletto, 2021 [28]	Brazil	60 to 60	3 months—6 (4.5–5.0)	34/26 to 32/28	Salivary iron	TXRF

NR, not reported. CMIA, chemiluminescence microparticle immunoassay. AAS, atomic absorption spectrometry. ECLIA, electrochemiluminescence immunoassay. TXRF, total reflection X-ray fluorescence.

**Table 2 children-08-01034-t002:** Quality score of the studies.

First Author, Publication Year	Selection	Comparability	Exposure	Total Score
Watanabe, 2011 [32]	****	-	***	7
Shaoul, 2012 [31]	****	**	***	9
Hussein, 2013 [25]	****	-	***	7
Koppal, 2013 [16]	****	-	***	7
Schroth, 2013 [30]	****	*	***	8
de Oliveira Buche, 2016 [23]	****	*	***	8
de Oliveira Buche, 2017 [24]	****	-	***	7
Jayakumar, 2017 [26]	****	-	***	7
Shamsaddin, 2017 [3]	****	-	***	7
Rajkumaar, 2020 [29]	****	**	***	9
Kotian, 2021 [27]	****	-	***	7
Poletto, 2021 [28]	****	**	***	9

Each asterisk indicates one score. **Selection**: Is the case definition adequate? (one score), Representativeness of the cases (one score), Selection of Controls (one score), and Definition of Controls (one score). **Comparability**: Comparability of cases and controls on the basis of the design or analysis (two scores). **Exposure**: Ascertainment of exposure (one point), Same method of ascertainment for cases and controls (one score), and Non-Response rate (one score).

## Data Availability

No new data were created or analyzed in this study. Data sharing is not applicable to this article.

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
