# Peer review of "Evaluation of Serum and Salivary Iron and Ferritin Levels in Children with Dental Caries: A Meta-Analysis and Trial Sequential Analysis"

_children, 2021, doi:10.3390/children8111034_

Round 1
Reviewer 1 Report
Dear Authors,
Thank you for the effort that you put in your work. The topic is really interesting and the work is well structured and explained.
However, Discussion and Conclusions do not give a clear explanation on how the results can be a useful addition and contribution to literature and to the clinical practice.
You will find my other comments by clicking on the highlighted parts of the attached manuscript.

Author Response
We thank Reviewer #1 for their helpful and valuable comments, which helped us to improve the quality of the manuscript. Please find the detailed point-by-point-response attached as a separate file. Thank you again for all your kind efforts.

Reviewer 2 Report
Dear Authors,
Congratulations for your work. Dental caries still has a high prevalence in children and any effort in connecting this disease with other systemic disorders is important.
Author Response
We thank Reviewer #2 for their helpful and valuable comments, which helped us to improve the quality of the manuscript. Please find the detailed point-by-point-response attached as a separate file. Thank you again for all your kind efforts.

Reviewer 3 Report
The manuscript is well written.
I would recommend authors register the review with Prospero.
The purpose of the manuscript in the last paragraph of the Introduction should be rewritten.
I would recommend authors create a new table for the RISK of bias tool.
Reference No 23 is pilot study, please explain this aspect in the discussion.
Limitations are well explained.
Kindly add a paragraph on the strengths and clinical relevance of this SR and MA.
I appreciate author's efforts really great work.
Author Response
We thank Reviewer #3 for their helpful and valuable comments, which helped us to improve the quality of the manuscript. Please find the detailed point-by-point-response attached as a separate file. Thank you again for all your kind efforts.

Round 2
Reviewer 1 Report
Dear Authors,
I would like once again to congratulate you on your work and appreciate the effort that you put in the revision. As previoulsy said, topic is really interesting and the work is well structured and explainen, and I now believe that there was a remarkable improvement after the thourough and detailed revision, that would make it interesting for publication.